# Photocatalytic Activity of Sulfanyl Porphyrazine/Titanium Dioxide Nanocomposites in Degradation of Organic Pollutants

**DOI:** 10.3390/ma15207264

**Published:** 2022-10-18

**Authors:** Tomasz Koczorowski, Barbara Wicher, Rafal Krakowiak, Kinga Mylkie, Aleksandra Marusiak, Ewa Tykarska, Marta Ziegler-Borowska

**Affiliations:** 1Chair and Department of Chemical Technology of Drugs, Poznan University of Medical Sciences, Grunwaldzka 6, 60-780 Poznan, Poland; 2Faculty of Chemistry, Nicolaus Copernicus University in Torun, Gagarina 7, 87-100 Torun, Poland

**Keywords:** porphyrazine, titanium dioxide, photocatalysis, photodegradation, thermal analysis

## Abstract

Magnesium(II) sulfanyl porphyrazine with peripheral morpholinethoxy substituents was embedded on the surface of titanium(IV) dioxide nanoparticles. The obtained nanocomposites were characterized with the use of particle size and distribution (NTA analysis), electron microscopy (SEM), thermal analysis (TGA), FTIR–ATR spectroscopy, and X-ray powder diffraction (XRD). The measured particle size of the obtained material was 327.4 ± 15.5 nm. Analysis with XRD showed no visible changes in the crystallinity of the material after deposition of porphyrazine on the TiO_2_ surface. However, SEM images revealed noticeable changes in the morphology of the obtained hybrid material: higher aggregation and less ordered structure of the aggregates. The TGA analysis revealed the lost 3.6% (0.4 mg) of the mass of obtained material in the range 250–550 °C. In the FTIR–ATR analysis, C-H stretching vibratins in the range of 3000–2800 cm^−1^, originating from porphyrazine moieties, were detected. The photocatalytic applicability of the nanomaterial was assessed in photodegradation studies of methylene blue and bisphenol A as reference environmental pollutants. In addition, the photocatalytic degradation of carbamazepine with porphyrazine/TiO_2_ hybrids as photocatalysts was studied, accompanied by an HPLC chromatography assessment of photodegradation. In total, 43% of the initial concentration was achieved in the case of bisphenol A, after 4 h of irradiation, whereas 57% was achieved in the case of carbamazepine. In each photodegradation reaction, the activity of the obtained photocatalytic nanomaterial was proved with almost linear degradation. The photodegradation reaction rate constants were calculated, and revealed 5.75 × 10^−5^ s^−1^ for bisphenol A and 5.66 × 10^−5^ s^−1^ for carbamazepine.

## 1. Introduction

The growing amount of pollutants entering into the soil and water, resulting from the expansion of industry in developing countries, with the simultaneous lack of effective methods of utilization of these pollutants, is one of the main problems for protecting our planet’s environment. What is more, the rising domestic consumption of drugs and cosmetics also has an impact on the environmental pollution. These micropollutions are currently not effectively eliminated in wastewater treatment plants. Pharmaceuticals, preservatives, and pesticides are still in the water in amounts that are detrimental to other organisms [1,2,3,4,5]. For example, the selected active pharmaceutical ingredients include steroids, β-blockers, terbutaline, NSAIDs, paracetamol, doxepin, imipramine, sulfamethoxazole, and carbamazepine that were found in wastewaters. The significant presence of these substances in the environment results from their overconsumption and improper disposal [6,7].

Due to their stability, the water concentrations of some modern pharmaceuticals, dyes, preservatives, and other chemicals are rising. What is most important is that the standard methods of water remediation (physical and biological processes; i.e., activated sludge or biofiltration) cannot efficiently handle this issue [8,9,10,11]. For this reason, so much effort is directed at this problem; thus, new remediation strategies are needed. Some new remediation methods were already tested, such as nanofiltration, with various types of materials or oxidation of pollutants after irradiation with UV light with or without the addition of hydrogen peroxide as a source of oxygen [12,13,14,15]. An interesting issue is using some photocatalysts for the photoremediation of common pollutants. Among others, titanium(IV) oxide has been widely considered a potential photocatalyst in the photodegradation of organic compounds as a component of hybrid systems, mineral or surface-functionalized carriers [16,17,18]. Its unique feature relies on the photoactivity after irradiation with UV light, which results in the potential application of titanium(IV) oxide in the advanced oxidation processes (AOP)—a new technique for the photodegradation of pollutants [19,20,21,22]. However, UV light represents approx. 4% of the solar radiation spectrum and, for this reason, the usefulness of TiO_2_ is limited [23]. Some strategies of surface functionalization, such as doping with metals and non-metals, and coating with selected organic compounds, were executed to enhance the photocatalytic potential of titanium(IV) oxide and widen the light wavelength range towards visible light [21,22,23]. One of the most promising ideas is the surface modification of TiO_2_ nanoparticles with porphyrazines.

Porphyrazines (Pzs) constitute a distinct group of porphyrinoids. They are macrocyclic complexes with four pyrrolyl rings fused by azamethine bridges. Depending on the type of precursor, the two main subclasses of Pzs are amino and sulfanyl derivatives [24]. The peripherally functionalized sulfanyl Pzs are well soluble in common organic solvents, and possess both optical and electrochemical properties [25,26,27,28]. One of the most important applications of sulfanyl porphyrazines is their potential use in photodynamic therapy [29,30,31], despite their symmetrical derivatives having low singlet oxygen generation quantum yields [32]. Moreover, these complexes were already used in wastewater treatment [33].

This paper presents the fabrication, physicochemical characterization, and photocatalytic applications of a hybrid magnesium(II) sulfanyl porphyrazine/TiO_2_ nanosystem. This publication is the continuation, and a significant extension of, the preliminary research carried out previously by our group [34]. It focuses on the physicochemical characterization of the obtained photocatalyst and more advanced studies of the photodegradation of organic compounds—dye, phenolic derivative, and selected drug.

## 2. Materials and Methods

### 2.1. Materials

The reactions were performed on a Heidolph MR Hei-Tec, equipped with Radleys Heat-On heating mantle. The solvents (dichloromethane, methanol, 99.5% of purity) were obtained from commercial suppliers (TCI, Zwijndrecht, Belgium) and used without further purification unless otherwise stated. The P25 Aeroxide titanium(IV) oxide was purchased from Sigma-Aldrich, Taufkirchen, Germany. The other reagents (bisphenol A, methylene blue, and carbamazepine) were purchased from Merck, Darmstadt, Germany; TCI, Zwijndrecht, Belgium; and Fluorochem, Glossop, UK.

### 2.2. Preparation of Photocatalyst

The synthesized porphyrazine complex was deposited on P25 Aeroxide titanium(IV) oxide (TiO_2_) nanoparticles using the chemical deposition method [35]. In general, a porphyrazine in the amount of 5 mg was added to a dispersion of 100 mg P25 nanoparticles (sized approx. 21 nm) in 20 mL of dichloromethane:methanol mixture (1:1, *v*/*v*) and stirred for 72 h. After the reaction, the solvents were evaporated on a rotary evaporator. Next, the obtained hybrid material Pz@P25 was air dried for 24 h. The ratio of the macrocycle to the P25 TiO_2_ was 1:20 (*w*/*w*).

### 2.3. Size and Distribution Measurements

The hybrid material was subjected to nanoparticle size measurements using a Malvern Panalytical NanoSight LM10 instrument (Malvern Panalytical Ltd., Malvern, UK), equipped with a sCMOS camera and 405 nm laser. Nanoparticle tracking analysis provided the data acquisition and storage (NTA) 3.2 Dev Build 3.2.16 software (Malvern, UK). Throughout, the nanoparticles’ dispersions were diluted with water (1 mg in 1 mL) to obtain the operating range of nanoparticle concentration. The measurements were performed at 25.0 ± 0.1 °C and the syringe pump infusion rate was set to 50 µL/min.

### 2.4. Thermogravimetry (TGA)

Thermogravimetric analysis was performed using a TG 209 F3 Tarsus instrument (NETZSCH, Selb, Germany). The open corundum crucible was used for measurements. The analyzed samples were heated at 5 °C/min from 25 to 900 °C under a nitrogen atmosphere (flow rate, 30 mL/min).

### 2.5. X-ray Powder Diffraction

X-ray powder diffraction (XRD) experiments were performed with a Bruker AXS D2 Phaser diffractometer with CuKα radiation (λ = 1.54060 Å). The operating voltage and current were maintained at 30 kV and 10 mA, respectively. The samples were scanned from 20° to 85° 2θ. Selected higher-quality scans were made with a step size of 0.02°, with a counting rate of 2 s/step with the sample spinning. A 1 mm slit module was used during measurements.

### 2.6. SEM

HR SEM micrographs were captured with an FEI Quanta 3D FEG (FEI, Hillsboro, OR, USA), with an acceleration voltage of 20 kV.

### 2.7. FTIR–ATR Measurements

The FTIR–ATR spectra were recorded between 400 cm^−1^ and 4000 cm^−1^, with a res-olution set to 1.0 cm^−1^, using a Shimadzu IRTracer-100 spectrometer equipped with a QATR-10 single bounce—diamond extended range, controlled by LabSolution IR software.

### 2.8. Photocatalytic Studies

The photocatalytic studies were performed using a ThalesNano PhotoCube photochemical reactor (ThalesNano Inc., Budapest, Hungary) equipped with a magnetic stirrer. In each experiment, 10 mg of Pz@P25 photocatalysts was added to 10 mL of 1 mM solution of each analyte (bisphenol A, methylene blue, and carbamazepine) in a glass vial and placed in the photoreactor. The irradiation of white light (380–700 nm) correlated to a color temperature of 6500 K with 100% of intensity (5920 lm) was applied for 300 min for bisphenol A and carbamazepine, and for 10 min for methylene blue while stirring the sample. During irradiation, the samples were collected, filtered through Pureland 0.22 µm nylon filters, and subjected to analysis:

(i) in the case of methylene blue, the UV–Vis spectra were recorded every 2 min, with the use of an Ocean Optics USB2000+ spectrophotometer (Ocean Optics, Dunedin, FL, USA) within the range of 200–1000 nm; and

(ii) in the case of bisphenol A and carbamazepine, the HPLC chromatography analyses were performed every 60 min using the Shimadzu LC-40D chromatograph equipped with a UV–Vis detector and column oven (Shimadzu, Kyoto, Japan). The Eurospher II 100-5 C18 column (150 × 4.6 mm) was used (Knauer, Berlin, Germany). The analyses were performed in isocratic flow at ambient temperature. The specified conditions of the HPLC analysis are described in detail in chapter 3.3. Photocatalytic Studies.

## 3. Results and Discussion

### 3.1. Preparation of Photocatalyst

The targeted magnesium(II) porphyrazine complex (Figure 1) was synthesized according to the protocol performed in previous research of our group from the mercaptomaleonitrile precursor [s34]. The obtained macrocycle was purified with flash column chromatography and subjected to spectral and electrochemical characterization described earlier. Moreover, its photoactivity was assessed with good results by means of singlet oxygen generation and cytotoxicity studies [31].

Due to the promising results of the initial photocatalytic studies performed using 1,3-diphenylisobenzofurane as a reference compound, we decided to provide a more in-depth physicochemical characterization of the photocatalyst and further photocatalytic experiments with synthesized magnesium(II) porphyrazine to assess its potential ability in the photodegradation of common pollutants. The photocatalytic hybrid material, denoted as Pz@P25, was prepared by the physical embedding of synthesized porphyrazine on the surface of TiO_2_ nanoparticles (commercially available Aeroxide P25, consisting of a mixture of crystal phases of anatase and rutile with a ratio of 9:1, declared by Sigma-Aldrich, Germany). Following the previously described procedure, the solution of the targeted compound was added to a TiO_2_ suspension, sonicated, and mixed for 72 h, yielding Pz@P25. The resulting hybrid material contained 5% (*w*/*w*) of the macrocycle.

### 3.2. Physicochemical Characterization of Photocatalyst

The sizes and the distribution of the obtained nanomaterials were subjected to detailed analyses using a NanoSight LM10 instrument (sCMOS camera, 405 nm laser) equipped with a nanoparticle tracking analysis system. The diameters of the materials were assessed and compared with pure TiO_2_ nanoparticles. The measured particle size values indicate strong agglomeration of the hybrid nanoparticles—both bare P25 and Pz@P25 nanoparticles showed several fractions in the NTA analysis (Figure 2). The mean particle size of the Pz@P25 (327.4 ± 15.5 nm) was four times higher than the unmodified P25 (74.8 ± 7.7 nm). This could suggest that the deposition of the macrocycles strongly influences the titanium(IV) oxide nanoparticles. In addition, in both cases, the calculated polydispersity indices (according to the formula PDI = (SD/mean diameter)^2^) were below 0.2 (0.13 for Pz@P25 and 0.17 for bare P25), which indicated that the distribution of nanoparticles within the studied hybrid material is monodisperse. It also seems that the presence of sulfanyl porphyrazine on the surface of P25 nanoparticles hampers the electrostatic interactions between Pz@P25 nanoparticles and allows obtaining Pz@P25 of specific diameters.

Thermogravimetric analysis showed no mass loss for the commercial P25 sample up to 900 °C (Figure 3, green), demonstrating the high thermal stability of the used nanoparticles. This result contrasts with the TGA analysis reported earlier for a sample of pure anatase with 40 nm nanoparticles size, for which significant mass loss associated with water evaporation was observed before 120 °C [18]. In the range 250–550 °C, the Pz@P25 lost 3.6% (0.4 mg) of the mass (Figure 3). Along with mass variation, the color is transformed from dark blue to white, proving Pz evaporation from the surface of the nanoparticles. However, assuming the 100% yield of Pz distribution on TiO_2_, the mass loss should be 0.5 mg.

X-ray powder diffractograms obtained for the samples of commercial P25 and Pz@P25 show no differences. The reflections are characteristic of a mixture of anatase and rutile. In Figure 4, the intensity at 2θ = 22.8° can be assigned to none of the titanium dioxide phases and is denoted with a star. Due to this unidentified reflection, only rough semi-quantitative phase analysis using Diffrac.Eva (Bruker AXS, 2021 in DIFFRAC. EVA V6.0.0.7 Bruker AXS GmbH, Karlsruhe, Germany) could be performed.

Anatase was estimated as a major phase constituting 83.8% of the sample. The similarity of the diffractograms before and after Pz coating indicates that Pz does not influence the titanium dioxide phases’ crystalline structures. Thus, Pz is probably well dispersed at the surface of the nanoparticles. The absence of additional intensities from Pz or a bumps characteristic of an amorphous state very often observed for Pz is due to Pz’s low content in the sample; thus, being below the detection limit of the XRD method. In addition, the peaks’ profile does not change, implying the same crystallite size of P25 and Pz@P25 samples. To verify this assumption, the average crystallite size was estimated using the Scherrer equation [36]: (1)$$\begin{array}{c} D_{S}=\frac{0.9\lambda }{\beta cos\theta }\; \end{array}$$
where: *Ds*—average crystallite size, *λ*—wavelength, 1.5406 Å, *β*—full width at half maximum, and *θ*—Bragg angle. 

Due to the samples being a mixture of anatase and rutile, the calculations were performed based on two non-overlapping intensities, with a maximum of 2θ = 25.3° and 48.1° for anatase; and on one with a maximum of 2θ = 27.4° for rutile. As assumed, the average crystallite size is similar for P25 and Pz@P25; however, it is about 20.5 nm for the anatase phase and 34.5 nm for the rutile. However, these calculations should be handled with great caution due to the limited number of intensities used in the calculations.

The powder diffractogram was also registered after the thermogravimetric analysis. The diffractogram shown in Figure 4c reveals the transformation of the anatase phase to rutile. The peaks are thinner than those observed for the commercial sample, indicating a larger crystallite size. Indeed, the calculated crystallite average size is 40.5 nm. The nanoparticle size growth after heating is in good agreement with previous reports on anatase–rutile conversion [37,38].

HR SEM micrographs (Figure 5) reveal highly aggregated particles. The particles appear pseudospherical in shape without irregularities. Nanoparticle tracking measurements showed that, in water solutions, bare P25 TiO_2_ formed regular aggregates with their size being close to multitudes of the core particle size, whereas Pz@P25 formed much bigger, disorderly aggregates. Those observations can be repeated in SEM images for dry material. In the case of bare P25 TiO_2_, the structure of the aggregates is regular—aggregate grains are composed of a small number of bare TiO_2_ particles and are joined tightly together. On the other hand, Pz@P25 shows a much less ordered structure with much bigger aggregate grains, whose sizes are close to the ones observed in NTA; in essence, revealing a different morphology than the unmodified material. As such, it can be said that the modification of TiO_2_ nanoparticles with Pz had a significant impact on the material’s morphology.

The FTIR–ATR measurements of pure P25 nanoparticles and fabricated material Pz@P25 (Figure 6) revealed the presence of four signals at 2965, 2935, 2860, and 2805 cm^−1^, originating from C-H stretching vibrations of the macrocyclic compound, confirming the successful deposition of porphyrazine on the surface of titania P25 nanoparticles.

### 3.3. Photocatalytic Studies

Literature data show that surface-functionalized titania nanoparticles can decompose organic pollutants after irradiation with UV light. Photodegradation studies were based on literature reports [39]. However, the results of photodegradation studies of erythromycin using P25 TiO_2_ nanoparticles surface functionalized with zinc(II) phthalocyanine obtained by Vignesh et al. proved that visible light can also be used to activate the hybrid photocatalytic system [40]. The preliminary studies showed that solely using UV (365 nm) light can cause fast bleaching of the macrocyclic photosensitizer grafted on the titania surface due to the formation of hydroxyl radicals, which can destroy the porphyrinoid. On the other hand, based on the results obtained by Musial et al., no photodegradation of substrate (ibuprofen and naproxen) was observed when red light (665 nm) was used [18].

For this reason, we performed the photodegradation experiments with the use of extensive white light (380–700 nm) driven by an LED lamp in the commercially available photoreactor ThalesNano PhotoCube. The goal of our experiments was to determine the photocatalytic activity of the obtained Pz@P25 nanosystem in the degradation of selected organic pollutants in water. Three substrates were chosen: methylene blue (MB) as a representative of dye pollutants; bisphenol A (BPA) as a common phenolic compound present in water and soil; and carbamazepine (CBZ)—an active pharmaceutical ingredient with increased uptake in society and known resistance to remediation in wastewater treatment plants.

In the case of methylene blue, the UV–Vis spectrophotometer was employed to monitor the very fast photodegradation of this dye. The white light irradiation of a 1 mM solution of MB in water for 8 min was performed in a 10 mL glass vial, and the 1 mL of samples were taken into a 10 mm quartz cuvette every 2 min. The recorded UV–Vis spectra in time are presented in Figure 7A. The results revealed a significant decomposition of methylene blue within 8 min. The observed photodegradation was nearly linear (R^2^ = 0.9614) (Figure 7C). Due to the known photodegradation ability of MB in the absence of any photocatalysts (photolysis), we also performed control studies, which indicated a very slight decomposition of the dye when irradiated with white light without the Pz@P25 nanosystem (Figure 7B).

The photodegradation of two other organic compounds—bisphenol A and carbamazepine—was observed with the use of HPLC chromatography according to literature data [41,42]. Furthermore, 1 mM water solutions of each analyte were prepared and irradiated with white light in a 10 mL glass vial for 4 h. After every 60 min, the samples were taken and subjected to HPLC analysis. In the case of BPA, the chromatograms were performed in 99.8% methanol, with a flow rate of 0.25 mL/min and detection at 275 nm (Figure 8A). The carbamazepine samples were analyzed in a 1:1, *v*/*v* mixture of distilled water and 99.8% methanol, with a flow rate of 2 mL/min and detection at 285 nm (Figure 8B).

After 4 h of the photocatalysis of bisphenol A with the use of Pz@P25 photocatalysts, a decrease of BPA concentration to approx. 43% of the initial concentration was observed (Figure 9A). The near linear correlation of the photodegradation of BPA (R^2^ = 0.9683) was achieved (Figure 9B). Similar experiments with the use of carbamazepine revealed the degradation of CBZ to approx. 57% of the initial concentration, which is a slightly worse result than the photodegradation of BPA (Figure 9A); however, with a slightly better linearity observed (Figure 9B).

The kinetic parameters (degradation rate constants *k* of BPA and CBZ decay) were calculated for experiments where the fabricated Pz@P25 nanosystem was used and irradiated with white light. The photodegradation of both analytes was assigned as first-order kinetics and, for this reason, the degradation rate constant was calculated using the formula:(2)$$\begin{array}{c} \ln P_{t}=\ln P_{0}-k\times t\; \end{array}$$
where: *P_t_* stands for the surface area of the sample in time *t* [h] in the isothermal test, *P*_0_ is the surface area of the sample in time 0, and *k* [s^−1^] is the reaction rate constant. The calculations are presented in Table 1:

The obtained results showed that both the calculated photodegradation reaction rate constants are very similar for both analytes, which indirectly proves that the chemical structure of the degraded compound does not influence the reaction. The obtained results of the photodegradation studies were compared with literature data. However, the objectivity of such comparison suffers from the differences in the light sources, irradiation times, light dosages, initial concentrations of analytes, and photocatalyst loading used in the various photodegradation studies. Table 2 summarizes the collected data.

## 4. Conclusions

Sulfanyl magnesium(II) porphyrazine, with morpholine moieties, was embedded on the surface of commercially available TiO_2_ nanoparticles. The obtained material was fully characterized in terms of its particle size and distribution (by NTA analysis), loss of mass in thermogravimetric measurements, morphology (SEM images), and polymorphism (XRD analysis). What is more, the FTIR–ATR measurements were also performed. The calculation based on NTA analysis revealed the monodispersity of fabricated material in water. The TGA of Pz@P25 showed that 3.6% (0.4 mg) of the mass was lost at 550 °C, indicating that the porphyrazine was not completely deposited on the surface of titania. The XRD analysis revealed no changes in the crystal structure of TiO_2_ after deposition. However, the changes in morphology were observed in SEM images—Pz@P25 aggregates were more irregular and disordered as compared to bare P25. The photodegradation studies of methylene blue, bisphenol A, and carbamazepine, with the use of the obtained material, were performed in water. The photoreactor’s white LED light was employed to initiate the photocatalysis. The UV–Vis spectrophotometer was used to monitor the photodegradation of methylene blue, whereas HPLC studies were performed in the case of other analytes. The photodegradation reaction of MB was much faster (minutes) compared to bisphenol A and carbamazepine, where the irradiation was performed for 4 h to indicate the significant decrease in their concentration. In all cases, the photodegradation reaction was defined as almost linear.

Considering all the obtained results, it can be concluded that photocatalysts based on TiO_2_ nanoparticles and porphyrazine moiety as a representative of photoactive compounds can be considered as a promising tool for the photodegradation of common water pollutants, and could be used as a supplementary method in wastewater treatment plants.

## Figures and Tables

**Figure 1 materials-15-07264-f001:**
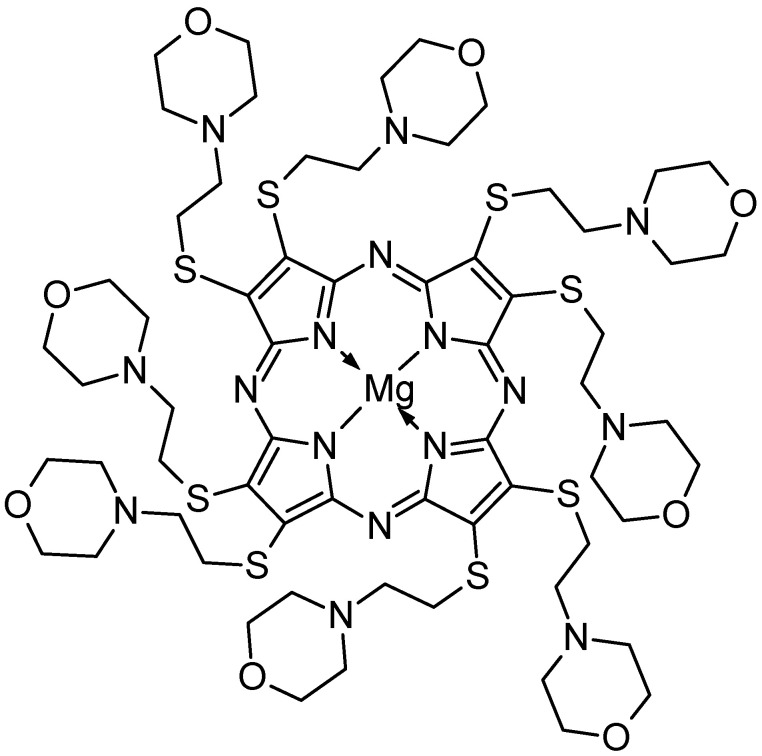
The chemical structure of the porphyrazine complex embedded on the surface of TiO_2_ nanoparticles.

**Figure 2 materials-15-07264-f002:**
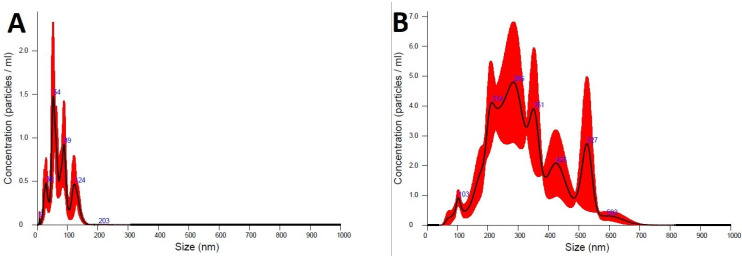
The particle size and distribution plots obtained in NTA analysis of bare P25 titania nanoparticles (**A**) and Pz@P25 nanoparticles (**B**).

**Figure 3 materials-15-07264-f003:**
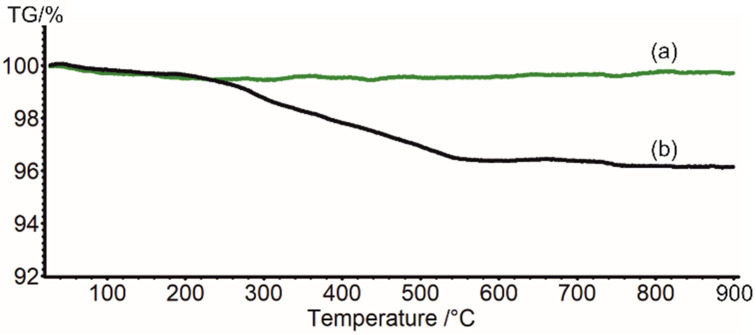
TGA curve of (**a**) commercial P25 and (**b**) Pz@P25.

**Figure 4 materials-15-07264-f004:**
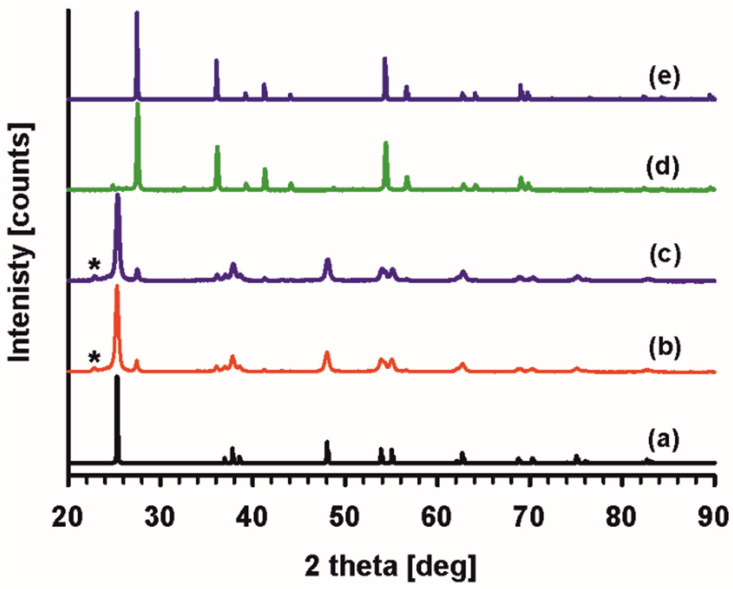
X-ray diffraction patterns of (**a**) anatase (PDF-03-065-5714), (**b**) P25, (**c**) Pz@P25, and (**d**) Pz@P25 after TGA experiment; and (**e**) rutile (PDF-01-070-4347). With *, the intensity that can neither be assigned to anatase nor the rutile be marked.

**Figure 5 materials-15-07264-f005:**
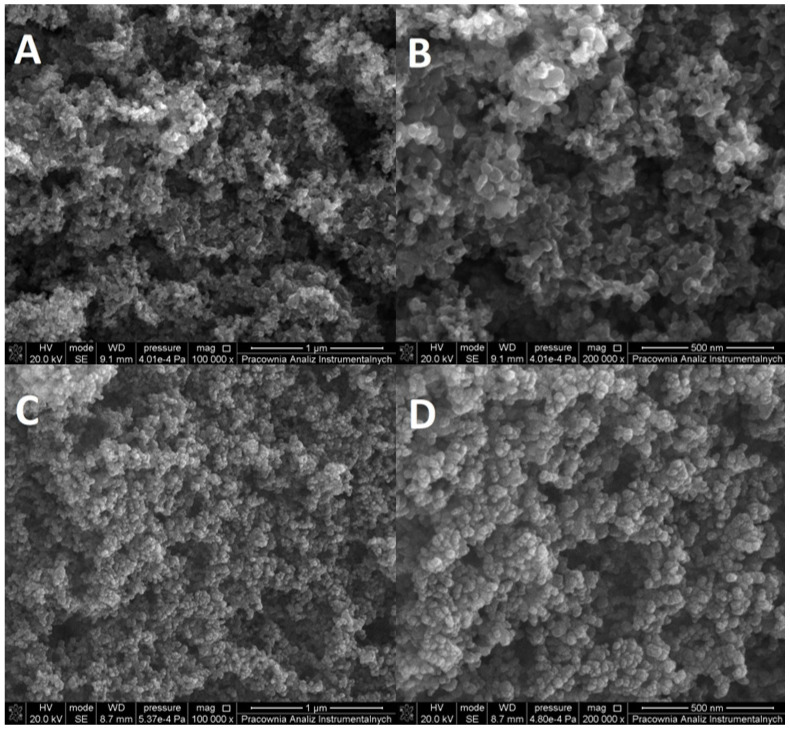
SEM images of Pz@P25 nanoparticles: (**A**)—magnitude 100,000×; (**B**)—magnitude 200,000×. SEM images of bare P25 TiO_2_ 21 nm nanoparticles: (**C**)—magnitude 100,000×; (**D**)—magnitude 200,000×.

**Figure 6 materials-15-07264-f006:**
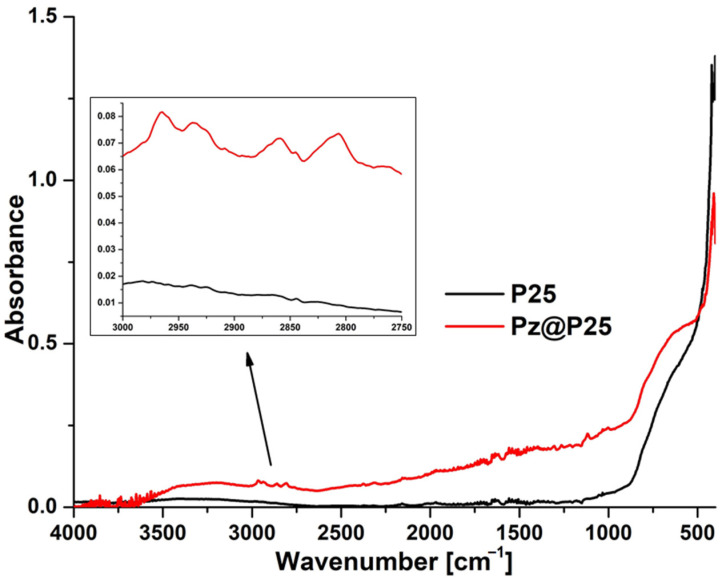
The FTIR–ATR spectrum of bare P25 (black line) and Pz@P25 (red line).

**Figure 7 materials-15-07264-f007:**
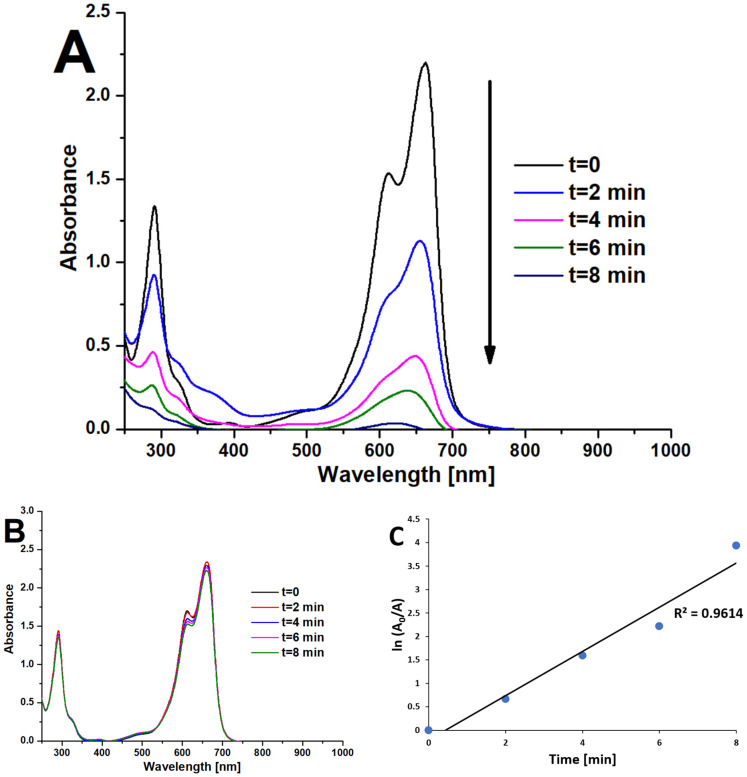
The UV–Vis spectra of methylene blue recorded during irradiation with white light in the presence of Pz@P25 (**A**) and the absence of photocatalyst (**B**). The logarithmic plot of methylene blue absorbance in time (**C**).

**Figure 8 materials-15-07264-f008:**
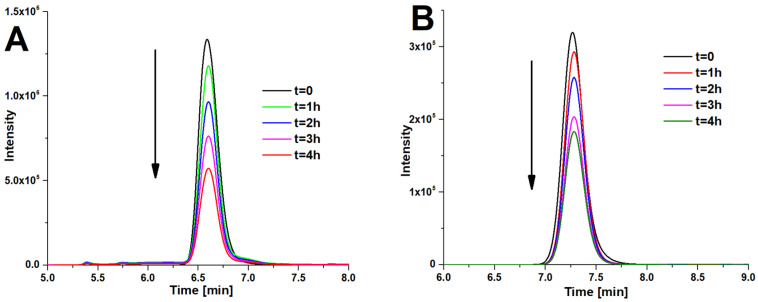
The HPLC chromatograms of the photodegradation of bisphenol A (**A**) and carbamazepine (**B**) in time.

**Figure 9 materials-15-07264-f009:**
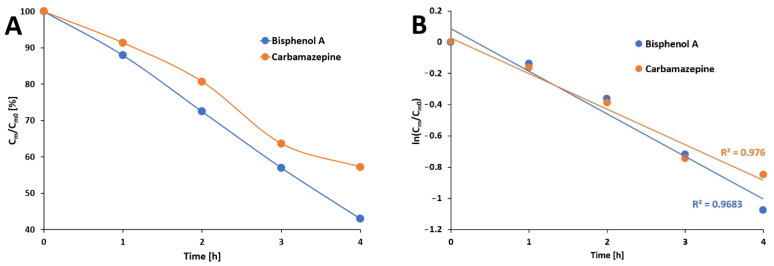
The changes in bisphenol A and carbamazepine concentrations after white light irradiation of the solution containing photocatalyst (**A**). The linear plot of photodegradation of targeted analytes (**B**).

**Table 1 materials-15-07264-t001:** The photodegradation reaction rate constants under white light irradiation of bisphenol A and carbamazepine.

	*k* [s^−1^]	Δ*k* [s^−1^]
**Bisphenol A**	5.75 × 10^−5^	1.44 × 10^−5^
**Carbamazepine**	5.66 × 10^−5^	1.42 × 10^−5^

**Table 2 materials-15-07264-t002:** The comparison of the literature data of diverse TiO2-based photocatalysts used in the photodegradation of bisphenol A and carbamazepine.

	Type of Photocatalytic Material	Photodegradation Efficiency [% of Initial Conc.]	Source of Light	Ref.
Bisphenol A	Bare P25 TiO_2_	7% after 4 h	UV lamp	[43]
	H_3_PW_12_O_40_/TiO_2_ composite	23% after 12 h	Xe lamp	[44]
	Bare P25 TiO_2_	68% after 1 h	UV lamp	[42]
	TiO_2_/C_3_N_4_ composite	0% after 0.5 h	Xe arc lamp	[45]
	Porphyrazine/P25 TiO_2_	43% after 4 h	White LED lamp	This work
Carbamazepine	Bare P90 TiO_2_	1% after 1.2 h	UV lamp	[46]
	TiO_2_/crumpled graphene oxide composite	0% after 2 h	UV + Xe arc lamps	[47]
	Hybrid MoS_2_/TiO_2_ systems	0% after 0.25 h	UV lamp	[48]
	Porphyrazine/P25 TiO_2_	57% after 4 h	White LED lamp	This work

## Data Availability

Not applicable.

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
