# Peer review of "Photocatalytic Activity of Sulfanyl Porphyrazine/Titanium Dioxide Nanocomposites in Degradation of Organic Pollutants"

_materials, 2022, doi:10.3390/ma15207264_

Round 1

Reviewer 1 Report

This manuscript has a certain degree of innovativeness, reliable data and particular research value. However, some problems and mistakes in this article need to be explained and corrected.

1.     The reaction solution cannot be filtered with a filter membrane after photodegradation for dye pollutants. Because of the adsorption of the dye by the filter membrane, the dye concentration will be significantly reduced.

2.     The authors need to supplement the necessary characterizations to illustrate the successful preparation of Pz@TiO2 composites, such as TEM, FT-IR, XPS, etc.

3.     The authors need to provide the degradation performance of pure TiO2 and Pz@TiO2 composites prepared by loading different amounts of Pz. To illustrate that the loading of Pz can enhance the photodegradation activity of TiO2.

4.     The optoelectronic properties are crucial to explaining the photocatalytic mechanism. The authors should perform the necessary UV-DRS, photocurrent and steady-state fluorescence tests to illustrate the enhanced photocatalytic performance.

5.     The active radical capture experiment is essential for the photocatalytic degradation reaction, and the author needs to supplement the corresponding experiment.

6.     Stability and reusability are essential indicators for investigating the practical application of catalysts. The authors need to study the cyclic degradation activity of the composites.

7.     The authors need to determine the energy band structure of the catalyst, clarify the transfer direction of photogenerated carriers, and propose a reasonable photocatalytic degradation mechanism.

8.     Suggestion: More recent papers involving photocatalytic environmental remediation can be referenced in the manuscript. (Liu C, Mao S, Wang H, et al. Chemical Engineering Journal, 2022, 430: 132806; Liu C, Mao S, Shi M, et al. Journal of Hazardous Materials, 2021, 420: 126613; Chemical Engineering Journal, 2022, 449: 137757).

Author Response

Dear Sir or Madam,

Please find enclosed our Response to the Reviewer.

Kind regards,

Tomasz Koczorowski

Reviewer 2 Report

This article describes the photocatalytic activity of sulfanyl porphyrazine/titanium dioxide nanocomposites. After careful characterization of the prepared nanocomposites, they were utilized for the degradation of organic pollutants. I suggest the manuscript for publication after considering the below points.

1.      What is the real novelty of this paper? Experimental design used in the present studies report the kind of experiments similar to most photocatalytic studies published previously. There is no novelty in experimental approach or any contribution to further mechanistic understanding.

2.      Abstract: provide some quantitative results/data

3.      The optical properties of sample should by analyzed and discussed.

4.      Band potentials are very important for photocatalytic reactions. The authors should measure VB and CB potentials using Mott- Schottky analysis.

5.      Line 85- ‘After the reaction the solvents were evaporated on a rotary evaporator’. What temperature was supplied to evaporate the solvents?

6.      Line 216- ‘The particles appear pseudospherical in shape without irregularities’. Authors are suggested to provide proper reason for this behavior with relevant literature.

7.      Why FTIR technique is not applied to confirm the material? Additionally, the authors are suggested to record FTIR of the materials before and after photodegradation to draw a proper reaction mechanism between the photocatalysts and pollutants.

8.       The findings in Fig. 3, TGA curve for commercial P25 (a) and Pz@P25  (b) should be compared vis-à-vis P25 and Pz@P25 after photodegradation of all three pollutants and inference must be drawn.

9.      Photoelectrochemical measurements should be conducted on the composites (at least, transient photocurrent responses, PL test).

10.  The authors should list the comparison between own composites and similar catalysts to show the innovation of the catalyst.

Author Response

(The authors gave the same response as above.)

Round 2

Reviewer 1 Report

Authors addressed very well most of my comments. Paper could be published now.

Reviewer 2 Report

All the questions asked by me have been well addressed excluding few instrumental analysis and the manuscript has been significantly improved by this major revision. The revised manuscript is good to get published.